# Improving Language Model Self-Correction Capability with Meta-Feedback

## Abstract

Large language models (LLMs) are capable of self-correcting their responses by generating feedback and refining the initial output. However, their performance may sometimes decline following self-correction, either because the feedback contains errors or due to unnecessarily attempting to refine an already accurate response. To address these limitations, we investigate whether the same LLM can generate *meta*-feedback that pinpoints errors in the feedback rather than the response, an ability that remains under-explored despite extensive research on LLMs' self-feedback generation. We design a novel self-correction prompting framework, **Feedback-on-Feedback (FoF)**, which leverages meta-feedback to improve the feedback before refining the response. Our framework first samples multiple pieces of feedback for the initial response, and prompts the LLM to generate a meta-feedback that analyzes the inconsistency between these feedback. Based on the meta-feedback, the LLM generates refined feedback that subsequently guides the revision of the response. Our FoF framework consistently outperforms competitive baselines across two LLMs on three datasets, covering arithmetic reasoning, machine translation, and programming tasks. Specifically, FoF improves performance on GSM8K by 3.6 points (45.2% vs. 41.6% for the initial answer) and on MBPP by 6.4 points (51.7% vs. 45.3%) using the LLaMA-3-8B model.

## 1 Introduction

LLMs have revolutionized the field of natural language processing, demonstrating exceptional performance across various tasks such as language generation, translation, and question answering (OpenAI et al., 2024). Despite their remarkable capabilities, LLMs often struggle with producing consistently accurate, coherent, and contextually relevant responses (Madaan et al., 2023; Chen et al., 2023b; Welleck et al., 2022). A critical area for improvement in LLMs is their intrinsic ability for self-correction—the capacity to identify and fix errors, inconsistencies, or shortcomings in their outputs without relying on external feedback, programs, or knowledge bases (Pan et al., 2023; Madaan et al., 2023; Chen et al., 2023b). This process typically involves the model first generating a critique that identifies the limitations of its initial response, followed by revising the response based on the self-generated critique. The critique-revise process can be iterated multiple times to progressively refine the model's output, allowing for a more thorough and comprehensive self-correction (Madaan et al., 2023; Shinn et al., 2023; Kim et al., 2023).

Many existing methods typically rely on external feedback or oracle labels (Madaan et al., 2023; Huang et al., 2024), which are often unavailable during inference. To address this, another line of research dives into the *intrinsic* self-correction ability (Huang et al., 2024) of LLMs to refine the answer without access to external information and oracle labels. However, they suggest that intrinsic self-correction harms the model performance in reasoning tasks since LM struggles to determine the correctness of the initial answer, leading to revising an answer that is already correct. Besides, the quality of LLM-generated feedback can be arbitrarily bad without proper guidance or selection, leading to inferior performance (Shridhar et al., 2023; Liang et al., 2023b). This motivates us to improve the quality of the feedback before applying it to refine the answer.

Particularly, LLMs' capacities to provide constructive feedback on their own self-feedback, known as meta-feedback (Lan et al., 2024), remains less explored. Lan et al. (2024) investigate this concept by prompting LLMs to critique the quality of their own feedback. While the study demonstrates

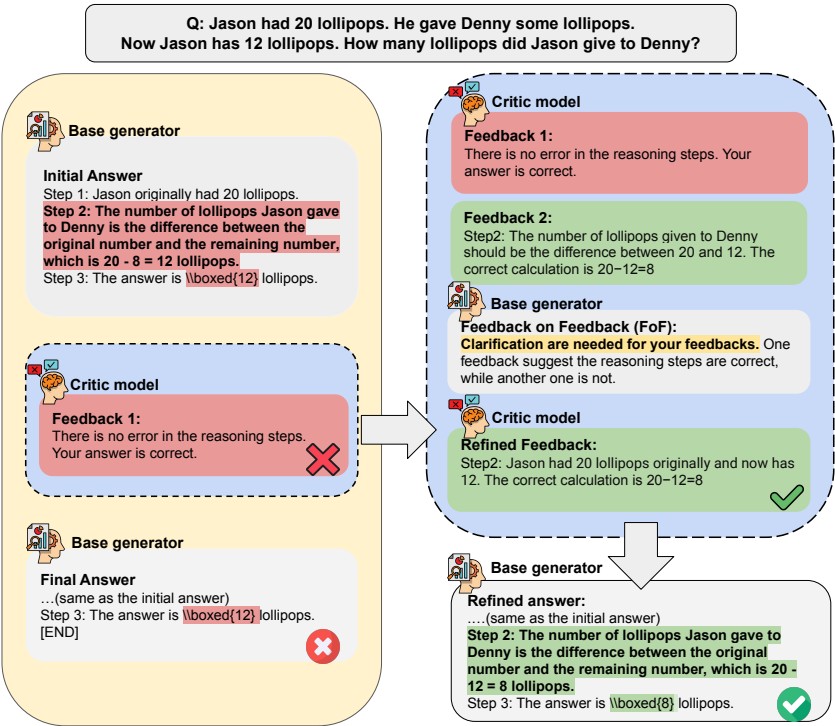

Figure 1: An illustrative example of FoF compared to the Self-refine setting (Madaan et al., 2023). The question is from GSM8K (Cobbe et al., 2021a), and all answers and feedback are generated by GPT-3.5-turbo-0515. In the Self-refine setting (left) , the base generator produces an initial answer, and the critic model provides feedback on it. However, since the feedback is incorrect, the answer remains a wrong answer . In the FoF setting (right) , two (or multiple) pieces of feedback responses are sampled from the critic model. The base generator recognizes the conflict between the feedback, prompting the critic model to clarify and correct it. Based on the refined feedback, the answer model updates the wrong answer to provide the correct answer .

that models like GPT-4 can generate meta-feedback, the findings also highlight significant limitations—LLMs struggle with consistency and accuracy of feedbcack, especially in complex tasks such as mathematical reasoning and coding. The quality of meta-feedback often lags behind human, indicating room for improvement in LLMs' abi9ility to self-evaluate. In this paper, we study the research question: **Can the meta-feedback improve the quality of feedback generated by LLMs, and subsequently enhance the final output?**

To this end, we propose a **F**eedback-**o**n-**F**eedback (FoF) framework. Inspired by self-consistency approaches, we focus on the consistency of self-feedback generated by LLMs. Specifically, we explore how identifying and resolving inconsistencies in self-generated feedback can improve the quality of the final output. Unlike methods that rely on external feedback or oracle labels, the FoF framework samples multiple feedback and then identifies inconsistencies between multiple LLM self-generated feedback based on their semantic similarities. Then, FoF 1) generates additional meta-feedback to analyze these inconsistencies, 2) refines the feedback with the meta-feedback, and 3) revises the answers using the refined feedback. An example of how FoF works is demonstrated in Figure 1. When the first feedback indicates the initial answer is correct and the second feedback shows there is still an error in the answer, combining different stances of feedback and the clarification from meta-feedback together provides more accurate feedback. This approach enables FoF to operate effectively in zero-shot scenarios without demonstrations, highlighting its generalizability across various tasks.

We conduct experiments on three datasets: GSM8K (arithmetic reasoning) (Cobbe et al., 2021a), CSMT (machine translation) (He et al., 2020), and MBPP (programming problem-solving) (Austin et al., 2021).Our FoF method outperforms the Self-Refine (Madaan et al., 2023), the Self-Consistency (Wang et al., 2023c) baseline and zero-shot CoT prompt (Wei et al., 2023) across all tasks and two models including one close-source model—GPT-3.5-0515 (Brown et al., 2020) and an open-source model—LLaMA3-8B (Touvron et al., 2023) in the zero-shot setting. Notably, FoF achieves an average improvement of 3.54 points across GSM8K, CSMT, and MBPP tasks compared to Self-Refine using LLaMA-3-8B. These results demonstrate the effectiveness of the FoF method in enhancing the self-correction ability of LLMs across various tasks and model sizes. Additionally, our ablation studies show that the quality of the critic model is key to the FoF framework's effectiveness. Using a more advanced critic like GPT-4 improves critiques and guides the base generator better.

Our contributions are threefold:

1. We introduce the **F**eedback-**o**n-**F**eedback (FoF) framework to enhance language model self-correction capability by aggregating over multiple pieces of self-feedback to generate more accurate feedback that guides answer revision.

2. We conduct experiments on GSM8K, CSMT, and MBPP using GPT-3.5-0515 and LLaMA-3-8B. The results demonstrate improvements of up to 6.4 percentage points on MBPP compared to the initial answer using the LLaMA-3-8B model.

3. We highlight the importance of selecting and integrating multiple feedback to improve answer accuracy. Our approach ensures more accurate and consistent self-correction by addressing inconsistencies between feedback.

## 2 RELATED WORKS

**Natural Language Feedback** The ability of LLMs to self-correct has garnered significant attention, with various approaches proposed to enhance this capability. Recent advancements leverage model natural language feedback and iterative refinement techniques (Ye et al., 2023; Madaan et al., 2023; Shinn et al., 2023; Kim et al., 2023). Approaches include iterative refinement through feedback alignment (Madaan et al., 2023; Gou et al., 2024; Ye et al., 2023), reinforcement learning for feedback optimization (Akyurek et al., 2023; Shinn et al., 2023; Kumar et al., 2024), and using external evaluation metrics to guide self-correction (Aggarwal et al., 2023; Paul et al., 2024; Zheng et al., 2023; Kim et al., 2023). Other methods integrate diverse prompts and verifiers, such as using self-verifiers or external verifiers to score reasoning paths (Gero et al., 2023; Li et al., 2023c; Zelikman et al., 2022; Cobbe et al., 2021b; Weng et al., 2023; Zhang et al., 2024), and multi-agent debate systems where LLMs interact to reach a consensus (Du et al., 2023; Cohen et al., 2023; Li et al., 2023a; Liang et al., 2023a). Notably, Kamoi et al. (2024b) highlights that LLMs can effectively self-correct under conditions such as task suitability, reliable feedback sources, model fine-tuning, strong self-evaluation mechanisms, and iterative feedback loops during inference, while Valmeekam et al. (2023) question LLMs' ability to self-critique effectively in planning tasks, further demonstrating the limitations of such frameworks.

However, some of the methods (Shinn et al., 2023; Madaan et al., 2023; Kim et al., 2023) depend on oracle labels or external feedback to determine when to stop the self-correction process. Multi-agent debate settings have also been found to be less efficient than self-consistency approaches (Huang et al., 2024). These issues and limitations raise questions about the true intrinsic self-correcting capabilities of LLMs (Huang et al., 2024). In contrast with those methods, our approach does not involve oracle labels and feedback from external verifiers. Our approach completely depends on the model's intrinsic self-feedback ability.

**Consistency in Reasoning Steps** Numerous types of research showcase that the accuracy of the final answer is influenced by the consistency of reasoning steps (Wang et al., 2023c; Li et al., 2023c). These approaches typically involve a "oversample-then-select" framework (Shridhar et al., 2023; Cobbe et al., 2021b; Weng et al., 2023), where methods like self-consistency sample multiple reasoning steps (Wang et al., 2023c) and then select the most consistent or reliable response, e.g. self-consistency samples the reasoning steps many times (Wang et al., 2023c), Adaptive Consistency which reduces sampling to 7.9 times with an early stop criterion, and SCREWS (Shridhar et al., 2023)

which integrates multiple selection methods like majority-voting and machine-selection. Confidence Matters (Li et al., 2024a) and Think Twice (Li et al., 2024b) sample answers and prompt the model to generate a new answer if conflicts arise between the first two responses. While all current works focus on the consistency on the reasoning steps, our method is crafted to focus on the consistency between feedback.

**Scaling Inference-Time Compute**    Recent studies have explored scaling inference-time compute to improve LLM performance (Brown et al., 2024; Snell et al., 2024). These approaches focus on increasing the number of answers or optimizing compute allocation to enhance output quality. In contrast, our approach introduces a new dimension by scaling feedback, not just answer outputs. This shift from sampling more answers to refining the quality and consistency of feedback expands the possibilities for improving model accuracy.

**Feedback Quality Evaluation**    Recent studies focus on evaluating the quality of feedback to enhance the self-correction ability of LLMs (Sun et al., 2024). Evaluating how well LLMs' outputs adhere to human values and ethical standards involves assessing biases, toxicity, and truthfulness (Hendrycks et al., 2023; Huang et al., 2023). Various approaches utilize both LLMs (OpenAI et al., 2024; Fu et al., 2023; Liu et al., 2023; Ke et al., 2023; Li et al., 2023b) and humans (Saunders et al., 2022; Wang et al., 2023b) as critics or annotators to evaluate and improve generated outputs. CriticBench (Lan et al., 2024) introduces a benchmark for assessing feedback and meta-feedback capabilities, emphasizing complex reasoning tasks and demonstrating that meta-feedback can significantly impact downstream performance. Recent works have proposed benchmarks to evaluate LLMs' ability to assess outputs: LLMBAR (Zeng et al., 2024) focuses on instruction-following, while ReaLMistake (Kamoi et al., 2024a) evaluates error detection across multiple categories. In contrast to the recent works, our method involves a analysis of feedback to detect and correct inconsistencies across multiple sampling, focusing on refining feedback before revising the original response. This approach provides a more targeted enhancement in feedback quality compared to existing benchmarks that center on instruction-following and error detection.

## 3   METHOD

In this section, we introduce the **Feedback-on-Feedback (FoF)** prompting method, which follows a three-step feedback refinement process: feedback generation, meta-feedback generation, and feedback refinement. A detailed FoF algorithm can be found in Appendix G. In this section, we first introduce the feedback generation and meta-feedback generation steps, and then we introduce the feedback refinement process.

**Base Generator**    The base generator is an LLM that takes the question $Q$ as input and generates an initial answer $R_0$. The initial answer is generated using zero-shot chain-of-thought prompting (Brown et al., 2020; Wei et al., 2023). Following (Madaan et al., 2023), we use the same generation prompt $p_{gen}$. Given an input question $Q$, a generation prompt $p_{gen}$, and a base generator $BG$, the initial answer $R_0$ is generated based on the combination of $p_{gen}$ and $Q$.

**Critic Model**    The critic model is another LLM that takes the CoT which contains the initial answer $R_0$ and the question $Q$ as input and provides feedback on the quality of the answer. To generate the feedback, we prompt the critic model with the prompt $p_{fb}$. All the prompts used for the critic are shown in the textbox below.

**Feedback Prompt:**
There is an error in the code above because of lack of understanding of the question. What is the error? To find the error, go through semantically complete reasoning steps and check if everything looks good.
**FoF Prompt:**
Disagree: Here are the two sampling feedback from the critic model on your previously generated reasoning step: Feedback sample 1: xxx. Feedback sample 2: xxx. The critic model is giving two different types of feedback, check the feedback and give the best feedback
Need Clarification = Here are the two sampling feedback...Clarifications are needed from the sampling feedback, try to clarify the feedback.
**Refined Feedback Prompt:** The programmer model may need clarification or disagree with you: FoF: xxx. Please give only one refined feedback based on the FoF from the programmer model. Your response should be similar to the previous round of feedback.

The critic model samples feedback with a temperature of 0.7 to generate $F_1$ and $F_2$ based on its training data and the given prompt.

**Feedback Refinement** The feedback refinement process aims to improve feedback quality and generate refined answers. It consists of the following steps:

FEEDBACK SIMILARITY CHECK We compute the semantic similarity $S$ between two feedback samples $F_1$ and $F_2$ using cosine similarity. We utilize the TF-IDF vectorization method (Jones, 2021) to transform the feedback samples into vectors. First, we apply TF-IDF to convert $F_1$ and $F_2$ into numerical representations, denoted as F1_vector and F2_vector. Following the vectorization, we calculate the cosine similarity between these two vectors using the formula:

$$S = \frac{\text{F1\_vector} \cdot \text{F2\_vector}}{\|\text{F1\_vector}\| \|\text{F2\_vector}\|} \quad (1)$$

This allows us to quantify the similarity between the two feedback samples based on their vector representations. The semantic similarity thresholds $\theta_1$ and $\theta_2$, are set at 0.5 and 0.8 respectively. These thresholds were chosen based on manual inspection of a few examples from the validation set. Based on these thresholds, we categorize the feedback similarity levels as follows:

- If $0 \leq S < \theta_1$, the feedback samples are considered to disagree with each other.

- If $\theta_1 \leq S \leq \theta_2$, the feedback samples need clarification, examples could be found in Section 5.3.

- If $S > \theta_2$, the feedback samples are considered to agree with each other.

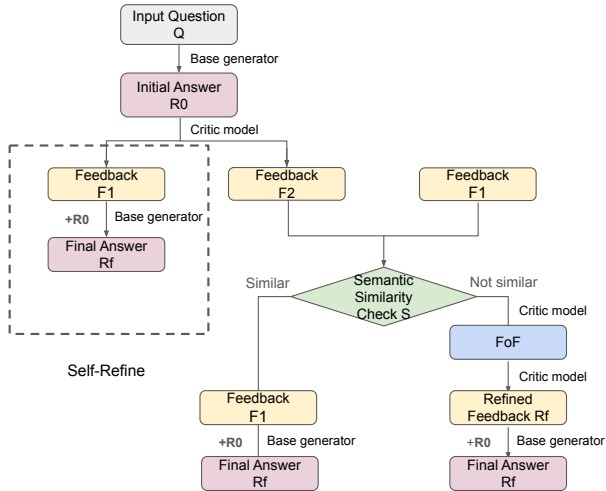

Figure 2: Block Diagram of the FoF method and the Self-Refine baseline. The left-hand side illustrates the Self-Refine method (Madaan et al., 2023), where the model generates feedback on its initial answer $R_0$, refines the answer using the feedback and $R_0$, and produces a final refined answer $R_f$. The right-hand side demonstrates the FoF method, which samples two feedback responses $F_1$ and $F_2$, comparing them through a semantic similarity check, and generating a fof if inconsistencies are found. The refined answer $R_f$ is generated based on the aggregated feedback and $R_0$.

FEEDBACK-ON-FEEDBACK (FoF) GENERATION    If the feedback samples $F_1$ and $F_2$ have low similarity, we generate FoF using the base generator $BG$ and the prompt $p_{fof}$.

REFINED FEEDBACK GENERATION    The refined feedback $RF$ is generated by the critic model $CM$ using all the history contexts including the question $Q$, the initial answer $R_0$, the FoF, and the feedback samples $F_1$ and $F_2$, and the prompt $p_{rf}$.

**Final Answer Refinement**    The final refined answer $R_f$ is generated by the base generator $BG$ using the question $Q$, the initial answer $R_0$, and the refined feedback $RF$, along with the refined answer prompt $p_{ra}$.

The refined answer $R_f$ is the final output of the FoF prompting method, which incorporates the feedback and refinement process to improve the accuracy and reliability of the generated answer.

## 4    EXPERIMENTS

**Models**    We utilize state-of-the-art language models as the base generator and critic in our FoF framework. We evaluate two LLMs, GPT-3.5-turbo and LLaMA3-8B, as our base models. We utilize the LLaMA3-8B model, which balances advanced capabilities with computational efficiency. Since GPT4 is considered as a strong model due to its performance on various benchmarks (OpenAI et al., 2024), we show the usage of GPT-4 as critic, showing that higher-quality feedback from a strong model can enhance accuracy, without any additional model training.

**Benchmarks**    We evaluate the performance of our FoF approach on three benchmarks requiring various reasoning skills. These evaluations span multiple types of tasks, covering arithmetic reasoning, commonsense reasoning, and programming problem-solving:

MATH REASONING:    We use the GSM8K dataset (Cobbe et al., 2021a), comprising 8.5K grade school math word problems to assess multi-step reasoning and numerical accuracy. For our evaluation, we specifically utilize the test set from GSM8K, which contains 1,319 examples.

MACHINE TRANSLATION:    We employ the Commonsense Machine Translation (CSMT) dataset (He et al., 2020) to evaluate translation quality using automatic metrics BLEURT (Sellam et al., 2020) and COMET (Stewart et al., 2020). BLEURT is a learned evaluation metric based on BERT, focusing on fluency and the extent to which the candidate conveys the meaning of the reference. COMET, on the other hand, is a neural framework that uses source text along with gold translations to measure both fluency and semantic accuracy. We take the test set from CSMT, which contains 200 examples.

PROGRAMMING PROBLEM SOLVING:    We use the MBPP (Multiple Benchmark Programming Problems) dataset (Austin et al., 2021), featuring 974 Python problems to test the model's ability to generate correct code given task description as input. We perform experiments on the test set of MBPP, which contains 500 python problems, where each problem has 3 unit tests. We follow prior work in including the first unit test in the prompt as part of the problem description (Chen et al., 2023b; 2021), and keep the remaining 2 unit tests hidden for a full evaluation. We evaluate MBPP based on the pass@1 metric, which indicates whether the single generated solution is correct (Chen et al., 2021).

**Prompt Selection Process**    Since LLMs are known to be sensitive to different prompts (Huang et al., 2024; Li et al., 2024a), to evaluate the impact of different feedback prompts on model performance, we experiment with several prompts inspired by related works (Huang et al., 2024). Appendix B presents the results of FoF using various prompts for the test sets of GSM8K and MBPP datasets. We found that the variance between prompts did not significantly affect the final results, as scores for the GPT-3.5-0515 model were relatively consistent, ranging from 74.22 to 79.22, and for the LLaMA-3-8B model, the scores range from 45.17 to 46.92, indicating some variability but not a drastic impact on overall performance. We use the same prompt for both FoF and Self-Refine (Madaan et al., 2023) to ensure a fair comparison.

**Baselines**    This section provides an overview of the baseline methods, including:

COT-PROMPTING: Chain-of-Thought (CoT) prompting (Wei et al., 2023) is a technique that elicits reasoning in large language models by encouraging them to generate intermediate reasoning steps before arriving at the final answer. This method enhances the model's ability to solve complex problems by breaking down the problem-solving process into smaller steps. The prompt typically contains instructions such as "let's think step by step".

SELF-REFINE PROMPTING: The primary baseline method in this study is the Self-Refine method (Madaan et al., 2023). Self-refine prompting is an iterative refinement method where the model generates self-feedback and uses it to improve its initial outputs. Huang et al. (2024) refer to this as critical prompting, which includes instructions like "find the error in your reasoning step". To ensure fairness between the Self-Refine and FoF settings, both methods start with the same initial answer and feedback round.

## 4.1 FEEDBACK SAMPLING

In our experiments, since we need to sample multiple feedback to generate meta-feedback, we sample two feedback responses from the critic model due to the context limit of GPT-3.5-Turbo with a temperature of 0.7. This temperature value ensures that the generated feedback samples are diverse (Renze & Guven, 2024; Wang et al., 2020; 2023a), allowing us to test the core idea of generating meta-feedback effectively.

## 4.2 STOP CONDITION

We follow the setup by Self-Refine (Madaan et al., 2023), where the feedback refinement process stops when it reaches the feedback round limit, or when the feedback contains the phrase "there is no error".

## 5 RESULT

Table 1: Performance comparison of different feedback methods across various models and datasets. Results are averaged over 3 runs with temperature=0.7, maintaining feedback randomness.

| | | GSM8K | | CSMT | | MBPP |
|---|---|---|---|---|---|---|
| | | Acc | Oracle Acc | BLEURT | COMET | Acc |
| GPT-3.5-0515 | + Initial Answer | $77.9_{\pm1.3}$ | $77.9_{\pm1.8}$ | $63.8_{\pm3.1}$ | $71.5_{\pm1.5}$ | $71.5_{\pm2.7}$ |
| | + Self-Consistency@10 | $78.3_{\pm2.4}$ | - | - | - | $74.5_{\pm1.6}$ |
| | + Self-refine | $77.4_{\pm1.7}$ | $78.8_{\pm1.9}$ | $66.1_{\pm0.8}$ | $74.1_{\pm2.3}$ | $74.1_{\pm0.5}$ |
| | + FoF | $78.7_{\pm2.0}$ | $80.1_{\pm1.2}$ | $67.4_{\pm2.1}$ | $75.3_{\pm1.7}$ | $75.3_{\pm2.3}$ |
| Llama-3-8B | + Initial Answer | $41.6_{\pm1.4}$ | $41.6_{\pm2.5}$ | $60.3_{\pm1.1}$ | $62.5_{\pm2.0}$ | $45.3_{\pm1.7}$ |
| | + Self-Consistency@10 | $42.2_{\pm3.2}$ | - | - | - | $45.4_{\pm1.9}$ |
| | + Self-refine | $43.5_{\pm1.0}$ | $44.0_{\pm2.8}$ | $63.1_{\pm1.7}$ | $66.0_{\pm2.4}$ | $49.1_{\pm0.9}$ |
| | + FoF | $45.2_{\pm1.9}$ | $45.7_{\pm1.4}$ | $66.3_{\pm2.7}$ | $68.0_{\pm3.1}$ | $51.7_{\pm2.4}$ |

## 5.1 MAIN RESULT

We perform evaluations using two different large-scale models across three benchmark datasets. As summarized in Table 1, the FoF method consistently demonstrates improvements across all benchmarks compared to the standard prompt and Self-Refine. For instance, using GPT-3.5-0515, our FoF method achieved an average accuracy of 78.71% on GSM8K, representing a 0.79% improvement over the standard prompt and a slight increase compared to Self-Refine. It is notable that the performance of GPT-3.5-0515 on GSM8K decreases after applying Self-Refine, this is aligned with the finding of (Huang et al., 2024). In some cases, Self-Refine even led to a decline in accuracy due to errors in the feedback. Our method addresses this issue by enhancing the quality of feedback through meta-feedback, which subsequently improves the final accuracy. Notably, for the LLaMA3-8B model, the FoF method achieve 45.17% accuracy, marking a 3.58% improvement over the standard prompt and a 1.68% increase compared to Self-Refine. The improvements from our method tend to decrease

as the model capability increases, yet the decision refinement stage consistently enhances performance across all models. It is notable that the performance of GPT-3.5-0515 on GSM8K decreases after applying Self-Refine, this is aligned with the finding of Huang et al. (2024). In the MBPP task, we assessed the effectiveness of the FoF method using the GPT-3.5-0515 and LLaMA3-8B models. As shown in Table 1, the FoF method achieved an accuracy of 75.27% with GPT-3.5-0515, reflecting a 3.77% improvement over the standard prompt and a 1.19% increase compared to the Self-Refine method.

In the Machine Translation Tasks, we evaluate the performance using the BLEU and COMET metrics. Our FoF approach achieves significant improvements in both BLEU and COMET scores after 4 rounds of iterative refinement. The BLEU score increases from 63.77 to 67.37, while the COMET score improves from 71.5 to 75.27. These results demonstrate the effectiveness of the FoF mechanism in enhancing the quality of the generated translations via iterative feedback and refinement rounds.

**Higher Feedback Quality Leads to Better Answers** Figure 3 visualizes the correlation between feedback scores, provided by the GPT-3.5-0515 LLM judger and CriticBench prompts (Lan et al., 2024), and answer accuracies for both methods. The FoF heatmap reveals a strong positive correlation, with 72% of data points falling into the high feedback score and high answer accuracy quadrant. In contrast, the Self-Refine heatmap shows a weaker correlation, with data points more evenly distributed across all quadrants.

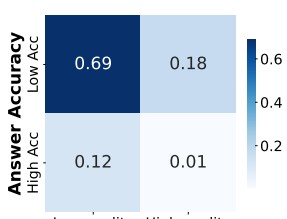
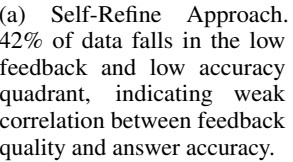
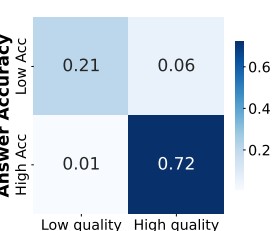

(a) Self-Refine Approach. 42% of data falls in the low feedback and low accuracy quadrant, indicating weak correlation between feedback quality and answer accuracy.

(b) FoF Approach. 72% of data falls in the high feedback and high accuracy quadrant, demonstrating a strong positive correlation.

Our analysis suggests that refining feedback through an iterative process improves the correlation between feedback scores and answer accuracy. This finding is consistent with CriticBench (Lan et al., 2024), which states that higher feedback quality leads to improved accuracy in question answering.

Figure 3: Heatmaps comparing the correlation between feedback score, which is prompted and calculated by the LLM, and answer accuracy for Self-Refine and FoF approaches. Note the imbalance: 228/500 examples have Self-Refine feedback, while 118/500 have FoF feedback. Despite fewer examples for FoF, higher feedback quality leads to higher accuracy in the MBPP task.

**FoF Changes More Answers Than Self-Refine** We further evaluate the changes in the answers after applying self-correction with the FoF method. The results on the GSM8K dataset using the GPT-3.5-0515 model show that our FoF method significantly increases the rate of Incorrect → Correct changes, demonstrating its effectiveness in enhancing answer accuracy. While both methods have similar percentages of wrong-to-wrong transitions (22.5% for FoF and 22.4% for Self-Refine), FoF outperforms Self-Refine in the wrong-to-correct category (3.2% vs. 3.5%) in each round. FoF generates more diverse answers than Self-Refine (Madaan et al., 2023) due to the additional meta-feedback stage, which encourages variability in response generation. This aligns with Huang et al. (2024), who note that mischanges from correct answer to incorrect result in self-correction failures. The improvements of FoF across tasks are due to fewer mischanges in feedback and answer rounds.

**Comparison with Self-Consistency** To ensure a fair comparison, we used a similar total number of tokens during inference between our FoF method and the self-consistency approach. Self-consistency involves generating 10 samples per iteration, while FoF involves one initial answer, three rounds of generation, two sampled feedback, one meta-feedback, and one refined answer, totaling 16 inference steps. Our results (Table 1) show that FoF consistently outperforms self-consistency across both GSM8K and MBPP datasets, with accuracy improvements ranging from 0.5% to 3%.

## 5.2 ABLATION STUDIES

**Critic Quality Matters** We conduct an ablation study to investigate the impact of the critic model's quality on the final performance of our FoF approach. We compared two critic models, GPT-3.5 and GPT-4, while keeping the base generator fixed as GPT-3.5. Table 2 presents the results on the GSM8K dataset. The findings highlight the importance of the critic model's quality in the FoF framework. By employing a more advanced language model as the critic, the system can generate higher-quality critiques, which in turn guide the base generator to produce more accurate corrections.

Table 2: Ablation study on the impact of critic model quality on final accuracy. Results are shown for the GSM8K dataset with GPT-3.5 as the base generator and using GPT-3.5, GPT-4 as the critic model.

| Base Model | Critic Model | Prompt Type | # of Feedback Samples | GSM8K Accuracy |
|---|---|---|---|---|
| GPT-3.5 | GPT-3.5 | + Standard Prompt | 0 | 77.27 |
|  |  | + Self-refine | 0 | 79.26 |
|  |  | + Self-refine | 2 | 77.78 |
|  |  | + FoF | 2 | **79.79** |
| GPT-3.5 | GPT-4 | + Standard Prompt | 0 | 78.24 |
|  |  | + Self-refine | 0 | 85.88 |
|  |  | + Self-refine | 2 | 85.48 |
|  |  | + FoF | 2 | **86.05** |

**Feedback Sampling Consistency** We also include a self-refine with two sampling variants to ensure a comparison using the same amount of API calls and a similar number of tokens. We sampled two feedbacks to not exceed the token limit of 4096. In this setting, self-refine generates two sampling feedback, and the base generator selects the one it has the most confidence in by using a prompt "Please compare the two pieces of feedback and choose the most appropriate one as the final feedback". The results indicate that LLMs lack the ability to choose the best feedback without external verification mechanisms, such as consistency checks. We further analyze the importance of feedback consistency by introducing a new baseline, USC on Feedback, which combines the Self-Refine approach (Madaan et al., 2023) with Universal Self-Consistency (USC) methods (Chen et al., 2023a). As highlighted in the appendix (Section A, the FoF method outperforms USC on Feedback by improving the accuracy on the GSM-Plus dataset (Li et al., 2024c) from 70% to 75%. The takeaway from these results is that incorporating consistency checks and leveraging multiple feedback samples can significantly enhance the performance of self-refinement methods in LLMs.

## 5.3 CASE STUDY

**Failed Examples and Analysis** While our FoF method demonstrates significant improvements in self-correction capabilities, it is important to acknowledge and analyze the instances where the method did not perform as expected. In this example, the initial answer is correct but includes incorrect intermediate steps. The two feedback samples provide incorrect guidance, leading FoF to agree with the inaccurate feedback and produce a refined response that reinforces the wrong answer. This case illustrates how multiple inaccurate feedback samples can compound errors, distracting FoF from identifying the correct solution. Consequently, FoF fails to recognize the initial answer as correct and follows the misleading feedback, resulting in an incorrect final answer. More failed and successful examples are available in Appendix D.5 and D.6.

---

**Gold Answer**: (...detailed reasoning steps) 7*2=14
**Initial Answer**: (...detailed reasoning steps) 5*(14/5) = 5*2.8 = 14 (*Correct initial answer with wrong middle steps*)
**Feedback sample 1**: The solution should round 14/5 (which equals 2.8) up to 3. (*Wrong feedback which focuses on fractions.*)
**Feedback sample 2**: There are 7 people in total so we should use 7*2 = 14. (*Correct feedback which leads to the correct answer*) **Clarifications are needed.**
**FoF**: The first feedback identifies the issue with dividing 14 by 5 and suggests rounding up, which results in 15. Based on the first feedback approach to the calculation, I agree that rounding up gives a more appropriate estimate. (*FoF agrees with the incorrect feedback*)

---

## 6 CONCLUSION AND FUTURE WORK

In this study, we investigate the FoF approach for enhancing the intrinsic self-correction (Huang et al., 2024) capability of LLMs by applying meta-feedback (Lan et al., 2024) in the feedback refinement process. Using our FoF method, LLMs achieve higher accuracy in math reasoning tasks and better quality in generation tasks and machine translation tasks. The study shows that the quality of the critic model is crucial in the FoF framework, as higher-quality critics generate better feedback, which positively correlates with improved performance and accuracy of the base generator.

In this work, we only explore sampling two pieces of feedback. Future work could extend this to multiple pieces of feedback to further enhance the feedback refinement process. Future work could explore integrating a reward mechanism into different stages of self-correction, such as reasoning steps, feedback, and meta-feedback, to guide the self-correction process more effectively (Yuan et al., 2024). Introducing a self-rewarding model that updates rewards during training could potentially overcome the limitations of treating all feedback equally and improve alignment with desired outcomes. Moreover, techniques such as multi-agent reasoning (Haji et al., 2024) and Constrained Chain-of-ToM (CCoToM) prompting (Lin et al., 2024) could further enhance the model's ability to understand and predict nuanced human intentions. Incorporating Logic-of-Thought (LoT) (Liu et al., 2024) to maintain logical consistency and integrating human-in-the-loop mechanisms (Cai et al., 2023) could refine the feedback process and improve model performance across diverse scenarios.

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

## A  EXTRA CONSISTENCY EXPERIMENT

Recent advancements have introduced the universal self-consistency (USC) method (Chen et al., 2023a), in which LLMs are prompted to select the most consistent response from multiple generated answers. To further emphasize the importance of feedback consistency, we introduce a new baseline, **USC on Feedback**, combining the Self-Refine approach (Madaan et al., 2023) with USC (Chen et al., 2023a). In this baseline, we first sample N pieces of feedback

Table 3: Results of GPT-4o-mini in GSMP-Plus

| Method | Accuracy on GSM-Plus |
|---|---|
| Self-Consistency on Answer | **0.76** |
| USC on Feedback | 0.70 |
| FoF | 0.75 |

and use the USC prompt to let LLMs select the most consistent feedback. This feedback is then used to refine the final answer. In order to make a fair comparison, we modify the FoF framework to operate under the same conditions as the USC on Feedback baseline. After generating N pieces of feedback, LLMs in the FoF approach identify inconsistencies and categorize feedback into three groups: Agree, Need Clarification, and Disagree. Based on this categorization, the framework proceeds with the usual FoF steps—generating refined feedback from the categorized responses, which is subsequently used to refine the answer. This experiment is conducted on the GSM-Plus dataset (Li et al., 2024c), using the cost-efficient and advanced GPT-4o-mini model (OpenAI, 2024). Since GPT-4o is trained based on GPT-4, and the GPT-4 training data includes GSM8K (Cobbe et al., 2021a), we opt to use GSM-Plus (Li et al., 2024c), an extended version of GSM8K that includes modifications such as numerical variation, arithmetic variation, problem rephrasing, distractor insertion, and critical thinking tasks. To maximize the use of the input token limit, we sample N=10 feedback in 1 round on 200 random shuffled example of GSM-Plus. As shown in Table 3, the FoF method improves the accuracy from 0.70 to 0.75 by selecting most consistent feedback.

## B  RESULTS OF PROMPT SELECTION

Table 4: Results of GPT-3.5-0515 and LLaMA-3-8B with different feedback prompts.

| Feedback Prompt in FoF | GPT-3.5-0515 | | LLaMA-3-8B | |
|---|---|---|---|---|
| | **GSM8K** | **MBPP** | **GSM8K** | **MBPP** |
| Assume that this answer could be either correct or incorrect. Review the answer carefully and report any serious problems you find. | 78.11 | 74.22 | 45.23 | 51.98 |
| Review your previous answer and determine whether it's correct. If wrong, find the problems with your answer. | 78.79 | 74.49 | 46.09 | 52.53 |
| Verify whether your answer is correct, and provide an explanation. | **79.22** | **75.43** | **46.92** | **52.76** |
| There is an error in the code above because of lack of understanding of the question. What is the error? To find the error, go through semantically complete reasoning steps and check if everything looks good. **(Our Prompt)** | 78.71 | 75.27 | 45.17 | 51.67 |

## C  COMPARISON BETWEEN FoF AND OTHER EXISTING WORKS

s

|  | Iterative Answer | Automated Critique | Zero-shot | Consistency on Answer | Consistency on Feedback |
|---|---|---|---|---|---|
| Self-Refine  (Madaan et al., 2023) | ✓ | ✓ |  |  |  |
| Self-Verification  (Gero et al., 2023) |  | ✓ |  |  |  |
| Reflexion  (Shinn et al., 2023) | ✓ | ✓ |  |  |  |
| Self-Debug  (Chen et al., 2023b) |  | ✓ |  |  |  |
| SelFee  (Ye et al., 2023) | ✓ |  | ✓ |  |  |
| Critical Prompt  (Huang et al., 2024) | ✓ | ✓ | ✓ |  |  |
| Think Twice  (Li et al., 2024b) | ✓ | ✓ | ✓ | ✓ |  |
| Confidence Matters  (Li et al., 2024a) | ✓ | ✓ | ✓ | ✓ |  |
| FoF (Ours) | ✓ | ✓ | ✓ | ✓ | ✓ |

Table 5: Summary of related work on intrinsic self-correction  (Huang et al., 2024) This table categorizes intrinsic self-correction methods based on five properties: Iterative Answer (the use of multiple rounds of answer generation), Automated Critique (whether the critique is generated automatically), Zero-shot (whether the method can perform without additional examples), Consistency on Answer (consistency in generating the correct answer), and Consistency on Feedback (consistency in the feedback provided).

## D  EXPERIMENT PROMPTS WITH DEMONSTRATION

Here are the shortened versions of the experimental prompts, including instructions and demonstrations. For clarity, only 1-2 demonstrations are provided per prompt. Each demonstrated result string is highlighted for readability, though this highlighting is not part of the actual prompt. Following each prompt, a test question and a cue (e.g., "Answer:") are provided to indicate where the result string should be placed.

### D.1  INITIAL ANSWER GENERATION

---

**FoF on GSM8K with the Initial Answer Round Shown Below**

=========================== system ===========================
Can you answer this question by writing reasoning steps and the answer:
On Monday, Walt walked 4 miles. Tuesday, he walked 6 times as many miles as he walked on Monday. His total mileage Monday through Wednesday was 41 miles. How many miles did he walk on Wednesday?
Please ensure your reasoning steps only consists of the essential steps and the final answer should be a single numerical number, in the form
boxedanswer, at the end of your response.
=========================== assistant ===========================
Reasoning steps: 1. On Monday, Walt walked 4 miles. 2. On Tuesday, he walked 6 times as many miles as he walked on Monday. Therefore, he walked 4 * 6 = 24 miles on Tuesday. 3. His total mileage Monday through Wednesday was 41 miles. So, he walked a total of 4 + 24 + x = 41 miles, where x represents the number of miles he walked on Wednesday. 4. Solving the equation, we get x = 41 - 4 - 24 = 13. Therefore, Walt walked 13 miles on Wednesday.
Answer: $\boxed{13}$ .

---

## D.2 FEEDBACK GENERATION

---

**FoF on GSM8K with 2 Sampled Feedback Shown Below**

=========================== assistant ===========================
There is an error in the code above because of lack of understanding of the question. What is the error? To find the error, go through semantically complete reasoning steps, and check if everything looks good
=========================== user ===========================
Here are the two sampling feedback from the critic model on your previously generated reasoning step:
The reasoning steps provided by the programmer model are incorrect. The mistake lies in step 4 where the equation is solved incorrectly. The correct equation should be 4 + 24 + x = 41, not 4 + 24 + x = 41 - 4 - 24. The reasoning steps provided by the programmer model are correct and logical. There are no logical issues with the steps to solve the math word problem. Clarifications are needed from the sampling feedback, try to clarify the feedback.

---

## D.3 FoF GENERATION

---

**FoF on GSM8K with Refined Feedback Shown Below**

=========================== assistant ===========================
Based on the two sampling feedback, there seems to be a disagreement about the correctness of the reasoning steps provided by the programmer model. One feedback suggests that there is a mistake in step 4, while the other feedback states that the reasoning steps are correct and logical.
To clarify the feedback, it would be helpful to understand the specific issue pointed out in step 4 by the first feedback. Additionally, it would be beneficial to know whether there are any other concerns or suggestions related to the reasoning steps provided.
=========================== user ===========================
Here is the refined feedback from the critic model.
The reasoning steps provided by the programmer model are correct and logical. There is no mistake in step 4 where the equation is solved.

---

## D.4 FINAL ANSWER GENERATION

---

**FoF on GSM8K with the Final Answer Round Shown Below**

=========================== user ===========================
Can you improve the reasoning steps in natural languages according to the feedback? Please ensure your final answer should be a single numerical number, in the form boxedanswer, at the end of your response.
=========================== assistant ===========================
Reasoning steps: 1. On Monday, Walt walked 4 miles. 2. On Tuesday, he walked 6 times as many miles as he walked on Monday. Therefore, he walked 4 * 6 = 24 miles on Tuesday. 3. His total mileage Monday through Wednesday was 41 miles. So, the combined distance he walked on Monday, Tuesday, and Wednesday is 4 + 24 + x = 41 miles, where x represents the number of miles he walked on Wednesday. 4. Solving the equation, we get x = 41 - 4 - 24 = 13.
Therefore, Walt walked 13 miles on Wednesday. Answer:
boxed13.

---

## D.5 FAILED EXAMPLES AND ANALYSIS

While our Feedback-on-Feedback (FoF) method demonstrates significant improvements in self-correction capabilities, it is important to acknowledge and analyze the instances where the method

did not perform as expected. Understanding these failures can provide valuable insights into the limitations of our approach and highlight areas for future improvement.

### D.5.1 FAILED EXAMPLE 1: ARITHMETIC REASONING

In one instance from the GSM8K dataset, the initial answer model generated the following response to a multi-step arithmetic problem:

Here are feedback from the critic model on your previously generated reasoning steps:

> **Feedback 1:** The error in the reasoning steps is the assumption that the total mileage for Monday through Wednesday is given as 41 miles. The prompt does not provide any information about the total mileage for Monday through Wednesday. Therefore, we cannot assume that the equation 7x + x + Wednesday's mileage = 41 is correct. The correct approach would be to set up the equation 7x + Wednesday's mileage = 41, but we would need additional information or a different equation to solve for Wednesday's mileage.

Despite iterative refinement, the final answer remained incorrect due to the conflicting nature of the feedback provided. This indicates that our method struggles when presented with contradictory critiques, especially in arithmetic reasoning tasks where precision is crucial.

> Let's assume that Walt walked x miles on Monday. On Tuesday, he walked 6 times as many miles as he walked on Monday, which is 6x miles. The total mileage for Monday through WednesSday is given as 41 miles. Let's represent Wednesday's mileage as y miles. So, the equation representing the total mileage would be x + 6x + y = 41. Simplifying the equation, we have 7x + y = 41. Since we don't have any other information, we cannot determine the exact values of x and y individually. Therefore, we cannot determine the specific number of miles Walt walked on Wednesday without additional information. **Answer:** ​Unknown​.

### D.5.2 FAILED EXAMPLE 2: MACHINE TRANSLATION

For the machine translation task using the CSMT dataset, the model was asked to translate the Chinese sentence "" into English. The initial translation was as follows:

> **Translation:** "He wants to recruit cadres from the same village to go underwater together and engage in drug trafficking."

The critic model's feedback included one response indicating that the translation was accurate, while another suggested an alternative translation to better capture the idiomatic meaning. The gold answer was:

> **Gold Answer:** "He wants to take the cadres of the same village to sell drugs with him."

This example highlights the difficulty in handling idiomatic expressions and the need for a more nuanced understanding of context during the feedback generation process.

### D.5.3 FAILED EXAMPLE 3: PROGRAMMING PROBLEM SOLVING

In the MBPP dataset, the model was tasked with solving a programming problem that required generating a function to find the volume of a triangular prism. The initial response was:

> *def find_Volume(base, height, length):*
>     *return base * height * length*

The critic model provided the following feedback:

> **Feedback:** There is no error in the code provided. The function 'find_Volume' correctly calculates the volume by multiplying the base, height, and length together.

However, the gold answer revealed that the function should calculate the volume of a triangular prism, which requires dividing the product by 2:

> **Gold Answer:** *def find_Volume(l,b,h):*
> *return ((l \* b \* h) / 2)*

The final solution did not incorporate the correct formula for calculating the volume of a triangular prism, resulting in an incorrect function. This failure illustrates the challenge of correctly interpreting and implementing mathematical formulas in code generation tasks.

### D.6 SUCCESSFUL EXAMPLES AND ANALYSIS

Despite multiple inaccurate feedback samples, a robust FoF process can identify and clarify the errors, leading to a correct refined feedback and final answer.

In this example, the initial answer is incorrect, and both feedback samples provide wrong suggestions. However, FoF successfully identifies the errors in both feedback samples, then provides a clear explanation of the correct formula, demonstrating its ability to identify and clarify errors in the feedback samples. This leads to a refined feedback that provides the correct answer, showcasing the robustness of FoF in handling inaccurate feedback and guiding the model towards the correct solution.

---

**Successful example due to correct FoF clarification**

**Gold Answer**:

```
def find_rect_num(n):
    return n * (n + 1)
```

**Initial Answer**:

```
def find_rect_num(n):
    return n * (n + 1) * (2 * n + 1) // 6
```

**Feedback sample 1**: The answer is using wrong formula, the correct formula should be (n \* (n + 1) // 2) \*\* 2. (*Wrong feedback*)
**Feedback sample 2**: The given answer complicates the problem; the answer should be n \*\* 2. (*Wrong feedback*)
Clarifications are needed from the sampling feedback, try to clarify the feedback.
**FoF**: The refined answers provided in both feedback are incorrect. Feedback sample 1 uses wrong formula. Feedback sample 2 oversimplifies the problem by suggesting n \*\* 2. The correct formula for the number of rectangles that can be formed using n points on a line is n \* (n + 1). (*FoF correctly identifies the errors in the feedback samples*)
**Refined Feedback**: The correct formula for the question should be n \* (n + 1) (*Correct Refined feedback*)

---

## E  PROMPT

## F  CRITICBENCH EVALUATION METHOD

## G  FoF ALGORITHM

```
Can you answer this question by writing
reasoning steps and the answer:{question}

Please ensure your reasoning steps only
consist of the essential steps and the
final answer should be a single numerical
number, in the form \\boxed{answer}, at the
end of your response.
```

(a) Initial prompt for generating the initial answer

```
Below, you'll find the reasoning steps from
the programmer model:{Reasoning Steps}

Assume the reasoning steps could be correct
or wrong.

Please go through semantically and
logically complete reasoning steps, check
if everything looks good and give your
feedback.
```

(b) Feedback prompt for generating $F_1$ and $F_2$

```
Here are the feedbacks from the critic
model on your previously generated
reasoning :{feedback}

Can you improve the reasoning steps in
natural languages according to the
feedbacks?

Please ensure your final answer should be
a single numerical number, in the form
\\boxed{{answer}}, at the end of your
response.
```

(c) Prompt used for feedback refinement

```
You are an experienced code reviewer responsible for
evaluating the feedback provided on a code snippet.
Please adhere to the following guidelines during your
review:

1. Begin by analyzing the accuracy and helpfulness of the
feedback in relation to the given code. Provide
constructive comments, and then affirm the feedback's
quality with a score from 1 to 10, where 1 denotes the
lowest quality and 10 denotes the highest quality.

2. You have access to the code snippet as well as the
feedback provided. However, you will not have access to the
improved code (if any) based on the feedback.

3. The feedback should correctly identify any errors or
areas of improvement in the code. If the feedback is
incorrect or not constructive, it should be given a low
score.
```

(d) Prompt used for scoring feedback quality

```
Need Clarification: Clarifications are
needed from the sampling feedbacks, try to
clarify the feedbacks

Disagree: Critic model is giving two
different feedbacks, check the feedbacks
and give the best feedback

Here are the two sampling feedbacks from
the critic model on your previously
generated reasoning step:
{Feedback Sample 1} + {Feedback Sample 2}
+ Need Clarification/Disagree
```

(e) Prompt used for generating FoF

```
The critic model may need clarification or
disagree with you:{fof_original}

Please give only one refined feedback based
on the fof from the critic model.
```

(f) Prompt used for feedback refinement

Figure 4: Collection of prompts used for various stages in the feedback generation and refinement process.

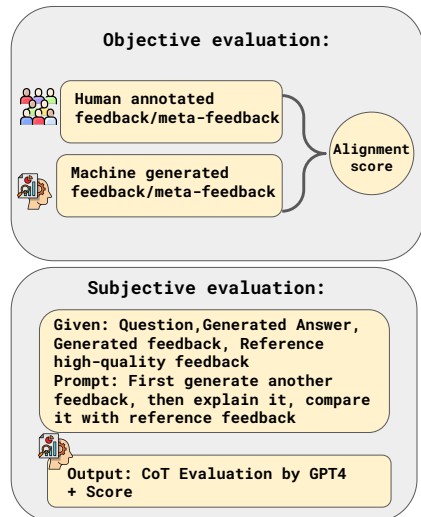

Figure 5: Overview of two evaluation methods in Criticbench

---

**Algorithm 1** FoF Algorithm

---

**Require:** Question $Q$, Base Generator $BG$, Critic Model $CM$, Semantic Similarity Thresholds $\theta_1, \theta_2$, Feedback Rounds

**Ensure:** Final Answer $R_f$

1: $R_0 \leftarrow BG(p_{gen} \parallel Q)$          ▷ Initial generation (Eqn. **??**)
2: **while** Round < Feedback Rounds **do**          ▷ Iterative refinement loop
3:     $F_1, F_2 \leftarrow CM(p_{fb} \parallel Q, R_0)$          ▷ Feedback generation (Eqn. **??**)
4:     $S \leftarrow \text{SemanticSimilarity}(F_1, F_2)$
5:     **if** $S < \theta_1$ or $\theta_1 < S < \theta_2$ **then** ▷ If feedback 1 and 2 disagree with each other or clarification needed
6:        $FoF \leftarrow BG(p_{fof} \parallel F_1, F_2)$          ▷ FoF generation (Eqn. **??**)
7:        $RF \leftarrow CM(p_{rf} \parallel Q, R_0, FoF, F_1, F_2)$          ▷ Refine feedback (Eqn. **??**)
8:     **else**
9:        $RF \leftarrow F_1$          ▷ Use first feedback
10:     **end if**
11:     $R_f \leftarrow BG(p_{fof} \parallel Q, R_0, RF)$          ▷ Refine initial answer (Eqn. **??**)
12:     **if** RF contains "this answer is correct" **then**          ▷ Check for stop condition
13:        **return** $R_f$
14:     **end if**
15:     $R_0 \leftarrow R_f$          ▷ Update initial answer for the next iteration
16:     $Round \leftarrow Round + 1$          ▷ Increment round counter
17: **end while**
18: **return** $R_f$          ▷ Return final answer after maximum rounds

---

