# OpenReview forum: "Improving Language Model Self-Correction Capability with Meta-Feedback"
_ICLR.cc/2025/Conference — ICLR 2025 Conference Withdrawn Submission_

### Official Review · Reviewer_wNBk · 2024-10-29

**Soundness:** 2
**Presentation:** 3
**Contribution:** 1
**Rating:** 3
**Confidence:** 2

**Summary:**

This paper proposed feedback-on-feedback (FoF) to provide meta-feedback on the feedbacks that the model generates for self-correction. The experiments are conducted on two LLMs (gpt-3.5 and llama-3-8B) and three tasks, math reasoning, machine translation and code generation. Results show that the proposed method yields 1-2 points of improvements over baseline methods such as self-refine and self-consistency.

**Strengths:**

* S1: Self-improving LLMs is an important and exciting domain to explore;
* S2: Strong baselines are used and compared against for the experiments, such as self-refine and self-consistency methods;
* S3: The analysis on the correlation between the feedback score and the answer accuracy is somewhat interesting

**Weaknesses:**

W1: I'm not sure if the method to identify similar feedback (as described in section 3 L239) is sound. If I understand correctly, here we use *TF-IDF* to measure cosine similarity, however, I don't think TF-IDF can identify semantic meanings of the sentences since it only matches words. (Also I don't think TF-IDF is from 2021 so the citation needs to be fixed);
W2: The performance improvements over previous methods, i.e., self-consistency and self-refine is not significant. More specifically, for most of the settings in Tab. 1, the difference is within the variance;
W3: (minor) Experiments are only conducted on GPT-3.5 and Llama-3-8B, it would be nice to see results on stronger models such as GPT-4 or Llama-3-405B, but this might due to the limited compute / query budget.

**Questions:**

* Q1: Have you tested the effectiveness of TF-IDF in identifying the similarity between the feedbacks for consistency? I can think of baselines such as BERTScore, or simply prompt a strong LLM (e.g., GPT-4) to see if the feedbacks are semantically similar.
* Q2: In the intro, it was mentioned that the models often can't provide good feedback, and the proposed method is to provide a meta-feedback on the feedback. Although I understand the self-consistency component might be the main driving force, I'm wondering what is the motivation to believe that it's easier to provide meta-feedback than the feedback itself?
* Q3: For the results in Tab. 1, what's the maximum refine steps allowed for self-refine? What about for FoF? And how many samples are drawn at each step?

---

### Official Review · Reviewer_wxkk · 2024-11-01

**Soundness:** 1
**Presentation:** 2
**Contribution:** 2
**Rating:** 3
**Confidence:** 4

**Summary:**

This paper proposes and presents "Feedback-on-Feedback (FoF)", an iterative prompting method that performs tasks with the following process: (1) input Q -> LLM -> prediction R0. (2) R0 -> LLM -> sample 2 feedbacks F1, F2. (3) if S(F1, F2) < \theta then (3a) F1, F2 -> LLM -> RF. else (3b) let RF be F1. (4) (Q, R0, RF) -> LLM -> Rf. The authors present a comparison of FoF against COT prompting, self-consistency, and self-refinement across 3 different datasets. Results suggest that FoF produces more helpful feedback than does self-refine, resulting in improved task performance.

RQ: Can meta-feedback improve the quality of feedback generated by LLMs, and subsequently enhance the final output?

**Strengths:**

* nice analysis showing that using feedback consistency in some symbolic way (in this case, cossim of tf-idf features) is better than prompting a model to select consistently
* some robust experimentation considerations, especially wrt keeping prompts consistent and holding number of tokens consistent
* figures explain the concepts well

**Weaknesses:**

## Results
The presentation of results should be improved. Section 5 does not present a compelling case for the main claim of FoF:  that since predictions can be improved with better feedback, we can get even better predictions by improving feedback with meta-feedback aided by symbolic consistency measures of sampled feedback.

Specifically:

Main results (overall results are not compelling):
* the performance improvement in Table 1 are not significant -- error margins overlap in almost all cases.
* ablations in Table 2 also do not show a notable improvement of FoF over self-refine (0 samples)

Section "FoF Changes More Answers Than Self-Refine" (ablations and analysis are unclear):
* Please include a table showing wrong-to-wrong, wrong-to-correct, correct-to-wrong counts for FoF-produced feedback vs. self-refine produced feedback. L418 and 419 have a few numbers, but they actually seem to suggest that self-refine is better at wrong-to-correct.
* L420-422 "FoF generates more diverse answers than Self-Refine ... which encourages variability in response generation.... The improvements of FoF across tasks are due to fewer mischanges in feedback and answer rounds." needs to be quantitatively validated: (1) metrics quantifying "FoF generates more diverse answers than Self-Refine", (2) metrics quantifying that "FoF [results in] fewer mischanges in feedback...", and (3) metrics quantifying that feedback from FoF results in "fewer mischanges in... answer rounds".

Section "Feedback Sampling Consistency":
* the table for this experiment actually shows that self-consistency compares to FoF

Would have liked an actual isolated evaluation of the first part of the RQ, that meta-feedback improves the quality of feedback generated by LLMs.

Sec 5.3 Case Study: the bottleneck problem with this example isn't specific to FoF right? Wouldn't self-refine also get this wrong? It would be nice to also see a case study that specifically is bad incorporation of meta-feedback or something.

## Nits and typos:
* L094 "abi9ility"
* Fig 3a has inconsistency between the caption (42%) and the heatmap (69%)
* L422 "...who note that mischanges from correct answers to incorrect result in self-correction failures." tautology? Did the authors mean to say something else?
* Fig 2 "Refined Feedback Rf" should be "Refined Feedback FR" ?

**Questions:**

* The baselines listed in section 4 are COT prompting and Self-refine prompting. There does not seem to be a clean mapping to the listed methods in the results table of section 5-- where are the COT results? What is "+ Initial Answer"? Can you include a description of all these baselines? (e.g. Self-Consistency is not described in experimental setup either)
* I like the consideration of number of tokens (L425-429)-- what were the actual resulting token counts?

---

### Official Review · Reviewer_CecZ · 2024-11-03

**Soundness:** 2
**Presentation:** 1
**Contribution:** 2
**Rating:** 3
**Confidence:** 4

**Summary:**

The paper proposes a prompting technique that provides feedback on the feedback generated by another paper. They show that it provides a minor improvement over the self-refine method on weaker models.

**Strengths:**

- Their proposed technique provides improvement over the self-refine on weaker models.
- They show stronger critic models such as GPT-4, provide better feedback, and also provide an example of where FOF fails.

**Weaknesses:**

- Although this method gets some improvement when using weaker model, but with a STRONGER critic model. Does this method provide any improvement with a stronger model (ex. GPT-4)?
- The proposed method makes several API calls compared to other methods, making the comparison unfair.
- Unnecessary complication of the methodology? The authors use semantic similarity to decide if the critic model should critic the feedback. However, I wonder why the LLM didn't just prompt the LLM to check the similarity and critique. Additionally, why did you use TF-IDF instead of using and ecoder model like BERT?
- The work presentation should be improved. For example, there is extended space below Figure 5.

**Questions:**

- Are the results in the table statistically significant?
- What are the improvements when using GPT-4 as the base model?
- In Table 2, why is the score for first row different for GPT3.5 base model. Shouldn't it be the same?
- In Figure 3 and its corresponding text, what is the feedback score, how do you calculate it?
- This work assumes that we have a stronger model for providing feedback. If we have a stronger model, then why should we not use it for generating response?
- Why don't the scores in Table 1 and Table 2 match for GSM8K?

---

### Official Review · Reviewer_a7v9 · 2024-11-04

**Soundness:** 3
**Presentation:** 3
**Contribution:** 2
**Rating:** 5
**Confidence:** 4

**Summary:**

This paper explores the idea of improving self refinement through feedback refinement as an additional step. Having noisy and hallucinated feedback is a known problems in refinement based approaches and techniques like self-consistency have tried to tackle this end to end. The proposed technique tries to improve the feedback independently.

**Strengths:**

- The insights especially around FoF changing more answers is very interesting.
- The technique does improve quality of generations.
- The idea to refine feedback is interesting.

**Weaknesses:**

- Only two models have been compared and the newer Omni models have been known to perform better on classical refinement.
- The improvement with FoF is not significant enough to understand if it is due to the new meta strategy or just another
refinement step would produce the same improvement.
- The premise of the paper is that "LLMs are bad at giving feedback" the proposed solution to this is using another round to get feedback on the generated feedback. It is not sufficiently justified in the paper, why the second feedback will have a lower hallucination and error rate. This needs to be justified for the paper to make intuitive sense.

**Questions:**

- Figure-2 exceeds bounding box.
- Figure 3 axis labels need to be more readable.
- Case study should have a few more examples for analysis or general insights.

---

### Note · Authors · 2024-12-02

I have read and agree with the venue's withdrawal policy on behalf of myself and my co-authors.